# Evolutionary Multi–Agent Reinforcement Learning for Crisis–Aware Demographic Policy Optimization

## Abstract

Demographic systems face unprecedented challenges from simultaneous crises. Conventional statistical demography techniques and agent–based models often struggle to capture nonlinear inter–regional interactions during periods of severe socio–economic disruption. To address this, we propose *MADDPG–EVO–DGM*, a hybrid algorithm that integrates multi–agent deep reinforcement learning with evolutionary optimisation and meta–learning principles to model regional demographic processes under multiple crisis scenarios. Each region is treated as an autonomous agent learning to steer demographic policy levers, while periodic evolutionary "boosters" overcome local optima via population–based perturbations of actor network parameters. Additionally, a Darwin–Gödel Machine–inspired meta–learning mechanism adapts the booster triggers, enabling self–improvement in the learning process. We evaluate MADDPG–EVO–DGM on a simulation environment calibrated with real demographic data for eight federal regions of the Russian Federation over the period 2000–2025 and subject to ten concurrent crisis scenarios (e.g., pandemic, geopolitical conflict, economic collapse). Experiments demonstrate significantly faster convergence and improved performance over a baseline MADDPG: the hybrid approach achieves a higher final average reward (252.57 vs 243.07) and $3.4 \times$ lower convergence variance ($\sigma = 0.24$ vs $0.80$), indicating more reliable training. It also exhibits qualitative performance jumps of $+68\%$ during evolutionary phases and maintains $35$–$45\%$ greater resilience under crisis shocks compared to the baseline. To our knowledge, this is the first application of multi–agent reinforcement learning to large–scale demographic modelling under crises, opening new possibilities for evidence–based, crisis–resilient population policy design. Code, data and logs are provided to ensure reproducibility.

## 1 Introduction

Demographic systems face unprecedented challenges from simultaneous crises. Conventional statistical demography techniques, such as the Lee–Carter model (Lee & Carter, 1992), and agent–based simulations (Billari & Prskawetz, 2003; Silverman et al., 2013) often struggle to capture nonlinear inter–regional interactions during periods of severe socio–economic disruption. The Russian Federation, with its 89 heterogeneous federal subjects, provides a compelling case: recent crises such as the COVID–19 pandemic (2020–2022), geopolitical instability with sanctions, and economic downturns have tested the resilience of regional populations in ways that static models cannot anticipate. This highlights a critical need for adaptive, data–driven approaches to demographic modelling under uncertainty.

Reinforcement learning (RL) offers a principled framework for dynamic decision–making in complex environments (Sutton & Barto, 2018). Multi–agent reinforcement learning (MARL) extends RL to systems of interacting decision–makers and is thus well–suited for modelling federated regions influencing each other's demographic outcomes (Zhang et al., 2021). However, straightforward application of MARL to demographic policy learning is difficult: the environment is non–stationary (policies of one region affect others), rewards are delayed due to the slow dynamics of population change, and naive gradient–based algorithms can become stuck in suboptimal equilibria. The MADDPG algorithm (Multi–Agent Deep Deterministic Policy Gradient) introduced by Lowe et al.

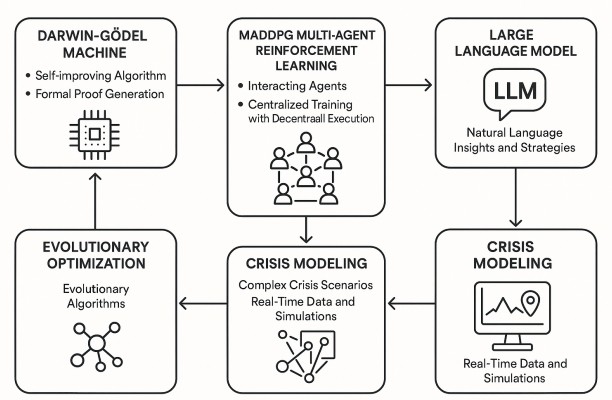

Figure 1: Conceptual overview of our hybrid approach integrating Darwin–Gödel machines, evolutionary optimisation, multi–agent reinforcement learning and crisis modelling. Real–time crisis data drive the agents and evolutionary boosters, while a large language model generates natural–language insights and strategies.

(2017) addresses some non–stationarity issues via centralized training with decentralized execution, yet gradient–based MARL can still converge to suboptimal policies—particularly under complex, multi–crisis dynamics—because it may get trapped in local optima.

In this work we introduce *MADDPG–EVO*, a hybrid MARL approach augmented with evolutionary search, and its extended variant *MADDPG–EVO–DGM* that incorporates meta–learning elements inspired by the Darwin–Gödel Machine concept (Zhang et al., 2025). Evolutionary algorithms provide a global, population–based exploration that complements gradient descent by escaping local optima and injecting diversity into policies. Periodic evolutionary boosters are integrated into MADDPG training: after a fixed number of episodes, the current actor weights are mutated to create a population of candidate policies; the candidates are evaluated and the best performer replaces the incumbent policy. This intermittent evolutionary injection draws inspiration from prior hybrid algorithms (Khadka & Tumer, 2018; Pourchot & Sigaud, 2019; Majumdar et al., 2020) but is applied here to demographic modelling for the first time. Moreover, our MADDPG–EVO–DGM variant adapts this process with a meta–learning strategy: instead of a fixed booster schedule, the system monitors learning progress and triggers evolutionary search whenever improvement stagnates, echoing the self–modifying loop of a Darwin–Gödel Machine (Schmidhuber, 2007; Zhang et al., 2025). This represents an initial integration of Darwin–Gödel Machine principles—open–ended evolution and self–improvement—into a practical MARL algorithm for socio–demographic systems.

Figure 1 summarises our proposed conceptual pipeline. A crisis modelling module produces complex crisis scenarios and real–time data streams which are fed into the MARL system. The large language model (LLM) component provides natural–language insights for scenario generation and parameter setting, while the evolutionary optimisation engine supplies population–based search to the Darwin–Gödel machine. These components together produce self–improving demographic policies that are evaluated and refined through simulations of demographic dynamics.

**Contributions.** Our contributions are fivefold:

1. **Crisis–aware demographic MARL.** We present the first application of MARL (MADDPG) to regional demographic and migration modelling at a country–wide scale, showing that agents can learn adaptive policies under multiple concurrent crisis scenarios.

2. **Evolutionary booster mechanism.** We integrate an evolutionary optimisation module into the MARL training loop to overcome local optima. The resulting MADDPG–EVO yields significant performance improvements and stability gains over standard MADDPG.

3. **Darwin–G"odel Machine augmentation.** We incorporate meta–learning principles inspired by the Darwin–G"odel Machine concept into the MARL framework—specifically

allowing the algorithm to modify its own training dynamics (booster timing) based on performance—thereby enhancing exploration and convergence reliability.

4. **Crisis environment with real data.** We construct a novel environment with ten distinct crisis scenarios (pandemic, war, economic collapse, climate disaster, energy crisis, etc.) parameterised by their impacts on demographic rates, integrated with real historical data for multiple regions. This enables quantitative evaluation of crisis modifiers on population dynamics across eight representative regions over 25 years.

5. **Empirical evaluation.** Through extensive experiments we demonstrate that our hybrid methods improve outcomes: early–training rewards more than double with evolutionary boosters, the final average reward increases by approximately 4%, and there is a $3.4\times$ reduction in performance variance. We analyse training phases (pre–and post–booster) and show reliable convergence and heterogeneous agent behaviours, highlighting policy implications for regional demographic management. We further compare against a pure evolutionary baseline and an independent learning baseline (see Appendix B.4) and confirm that both lag far behind our hybrid methods, underscoring the necessity of combining gradient learning with evolution.

Although we restrict our experiments to eight representative regions for computational feasibility, these regions were chosen to reflect a diverse cross–section of socio–economic profiles (urban, rural and industrial). The underlying algorithm and environment scale naturally to a larger number of agents via parameter sharing and parallel computation, albeit at increased computational cost.

**LLM usage disclosure.** We used a large language model (ChatGPT) to assist with language polishing, LaTeX troubleshooting, and limited research ideation (e.g., framing the comparison to population-based training and structuring ablations). The LLM did not generate or analyse data, did not implement algorithms, and did not produce experimental results. No automated web retrieval via the LLM was used; all references and claims were curated and verified by the authors.

## 2 RELATED WORK

**Demographic modelling under crises.** Traditional demographic forecasting methods (e.g., the Lee–Carter model (Lee & Carter, 1992)) rely on statistical extrapolation and often struggle with structural breaks during crises. Agent–based models have been employed to capture interactions and emergent phenomena in demographic systems (Billari & Prskawetz, 2003; Silverman et al., 2013), but such methods typically do not incorporate decision–making or multi–agent interactions. Recent studies have begun exploring machine–learning approaches for demographic modelling: Bohk–Ewald *et al.* (Bohk-Ewald et al., 2018) applied predictive modelling to mortality and fertility rates, but such methods typically do not incorporate policy optimisation. Our work differs by focusing on policy optimisation in a multi–region system using MARL to actively learn how regional governments could respond to crisis shocks rather than merely forecasting population metrics.

**Multi–Agent Reinforcement Learning.** MARL has advanced rapidly in domains such as games, robotics and traffic control, but its use in social systems modelling remains nascent. Algorithms like MADDPG (Lowe et al., 2017), COMA (Foerster et al., 2018) and QMIX (Rashid et al., 2018) address challenges such as non–stationarity and credit assignment. MADDPG in particular enables continuous action coordination via centralised critics and has shown effectiveness in mixed cooperative–competitive tasks. To our knowledge MARL has not previously been applied to computational demography. In our formulation, each region is a learning agent—conceptually related to agent–based models but with agents learning optimal strategies rather than following fixed rules. MARL allows agents to implicitly learn migration dynamics by optimising their own region's outcomes in the context of other agents' actions. Early work applying MARL to socio–economic settings (e.g., coordinating tax policies or economic games) is beginning to emerge, but these efforts are still limited in scope. Our work is among the first to bring MARL to the domain of demography.

**Hybrid evolutionary reinforcement learning.** There is growing evidence that combining evolutionary algorithms with RL can yield improved exploration and performance on challenging tasks. Khadka & Tumer (2018) and Majumdar et al. (2020) showed that evolutionary population search

can assist multi–agent coordination by providing a diverse set of experiences. Similarly, Pourchot & Sigaud (2019) introduced a genetic algorithm alongside policy gradients to escape local optima in continuous control tasks. These hybrid approaches exploit the complementary strengths of evolution (global search, diversity) and gradient descent (efficient fine–tuning). We integrate an intermittent evolutionary booster into MADDPG training for the first time in a demographic context. Unlike methods that run evolution continuously or in parallel, our booster monitors learning progress and triggers an evolutionary search whenever improvement stalls, injecting diversity into a slowly evolving environment. Furthermore, inspired by open–ended learning frameworks, we allow the booster schedule to adapt based on learning progress in our DGM–augmented variant. This aligns with the concept of a self–improving Darwin–G"odel Machine recently proposed by Zhang et al. (2025), which advocates an agent architecture capable of modifying its own algorithms through an evolutionary search process. Our approach takes a step in this direction by enabling the training algorithm itself to evolve (in terms of when and how it explores), not just the policy parameters.

**Comparison with population–based training.** While several *population–based training* (PBT) and auto–RL frameworks also combine evolutionary search with gradient learning, they operate quite differently from our method. PBT and related approaches such as CEM–RL or evolution strategies maintain multiple independent learner populations and periodically copy weights or hyperparameters between them on a fixed schedule to explore hyperparameter spaces (Jaderberg et al., 2017; Salimans et al., 2017). In contrast, our booster perturbs a *single* shared policy network across all agents and is *triggered adaptively* whenever the moving average of the reward stops improving. This design yields a more sample–efficient adaptation suited to cooperative multi–agent settings. To our knowledge evolutionary boosters have not previously been applied to MARL, particularly in the domain of demographic policy modelling.

## 3 METHODOLOGY

We model demographic dynamics across multiple regions as a multi–agent sequential decision process. The environment describes the crisis environment and data integration, the state and action spaces, the MADDPG architecture and training procedure, and the evolutionary booster mechanism.

### 3.1 CRISIS–AWARE DEMOGRAPHIC ENVIRONMENT

Our environment simulates population and migration dynamics across $N$ regions on annual time steps. For realism, baseline demographic trajectories are derived from historical data on population and socio–economic indicators for Russia's federal subjects. In this study we focus on a subset of $N = 8$ representative regions due to computational constraints, but the framework is conceptually applicable to all 89 regions. The simulation timeline covers years 2000 through 2025 (25 years), using the year 2000 data as an initial state. On top of the baseline demographic trends, we introduce *crisis modifiers* representing exogenous shocks to demographic rates. We define ten distinct crisis scenarios, each characterised by a name, duration, and percentage changes to key demographic parameters. For example, the COVID–19 pandemic (2020–2022) reduces the birth rate by 15 %, increases the death rate by 25 % and suppresses net migration by 60 %; a geopolitical conflict (2022–2024) reduces births by 20 % and increases deaths by 15 % while almost eliminating migration; an economic collapse (2021–2023) reduces births by 25 %, increases deaths by 10 %, reduces net migration and GDP per capita by 40 %; a climate disaster (2022–2027) reduces births by 10 %, increases deaths by 20 % and halves migration; and an energy crisis (2021–2024) reduces births by 18 %, increases deaths by 30 % and reduces migration by 70 %. Multiple crises can overlap, compounding their effects: if a region experiences both a pandemic and a geopolitical conflict in 2022, its birth rate is reduced by approximately 35 % (combining the two shocks) and migration is heavily suppressed. Crisis definitions (names, timelines, percentage impacts) are specified in an external scenario file (e.g., `crisis.txt`) and can be easily extended to new crisis types or different severity profiles.

The environment state $s_i^t$ for region $i$ at time $t$ includes current population and vital rates as well as economic indicators: we incorporate population, birth rate, death rate, net migration rate, GDP per capita, unemployment rate and average wage. Population evolves according to the demographic

balance equation

$$\text{population}_i(t+1) = \text{population}_i(t) + \text{births}_i^t - \text{deaths}_i^t + \text{migrants}_i^t, \qquad (1)$$

with births, deaths and migrants computed from the current state and adjusted by any active crisis modifiers. Notably, migration between regions couples the agents: the net migration gain/loss of region $i$ is affected by the relative attractiveness of other regions $j$ (e.g., regions with higher average wage or lower unemployment may draw migrants from $i$). We implement an inter–regional migration flow model such that total out–migration from region $i$ is distributed across destinations $j$ in proportion to each destination's attractiveness. This induces a coupling where improving conditions in one region can negatively impact others by drawing away population. Each simulation episode lasts 25 time steps (years), allowing long–term evaluation of policy effects up to year 2025 from a 2000 baseline.

### 3.2 AGENTS: ACTIONS, OBSERVATIONS AND REWARDS

Each region's government is modeled as an agent with a continuous action $a_i^t \in [-1, +1]$ representing an abstract policy lever that can influence demographic outcomes. Positive actions correspond to pro–population measures (e.g., family incentives, healthcare improvements, programmes to attract migrants), while negative actions correspond to disinvestment or policies that indirectly discourage population growth. The action modifies the region's vital rates: a high positive action increases net migration inflow and slightly boosts the birth rate, whereas a negative action has the opposite effect (e.g., causing out–migration or lowering incentives for family expansion). All agents act simultaneously at each time step, and the environment updates the state of all regions for the next year accordingly.

An agent's observation $o_i^t$ consists of its own state $s_i^t$ (local demographic and economic features) and binary indicators of which crises (if any) are currently active in that region. Agents do not observe other agents' states or actions directly—inter–regional effects are felt through the state dynamics (for example, a region might see increased out–migration if another region's policies make it more attractive). The reward for each agent is designed to reflect desirable demographic outcomes for its region. We use a per–region reward function balancing population growth, migration retention and economic stability:

$$r_i^t = w_{\text{pop}} \, \Delta\text{population}_i^t - w_{\text{mig}} \left| \text{out\_migration}_i^t \right| - w_{\text{unemp}} \, \Delta\text{unemployment}_i^t, \qquad (2)$$

where $\Delta\text{population}_i^t$ is the change in population during year $t$ (favouring growth or smaller declines), $\left| \text{out\_migration}_i^t \right|$ is the absolute number of people leaving region $i$ (penalising loss of population), and $\Delta\text{unemployment}_i^t$ is the change in unemployment rate (penalising economic decline). We set weights $w_{\text{pop}} = 1$, $w_{\text{mig}} = 0.5$ and $w_{\text{unemp}} = 0.2$ to balance the scales of these terms. Each agent seeks to maximise its discounted cumulative reward

$$\sum_{t=0}^{\infty} \gamma^t r_i^t, \qquad (3)$$

with discount factor $\gamma = 0.95$ capturing some foresight (approximately a 20–year horizon, given annual steps).

While the reward drives learning, we also evaluate a *stability metric* defined as the inverse coefficient of variation of the total population. This demography-inspired measure quantifies how steady the population remains over time; it is used solely for post-hoc analysis of policy behaviour and is *not* part of the agents' optimisation objective.

### 3.3 MADDPG ARCHITECTURE AND TRAINING

We adopt the MADDPG algorithm (Lowe et al., 2017), an extension of deterministic policy gradient methods to multi–agent settings. Each agent $i$ has an *actor* network $\pi_{\phi_i}$ that maps its observation $o_i^t$ to an action $a_i^t$, and a *critic* network $Q_{\theta_i}$ that estimates the Q–value for agent $i$ given the joint state and joint actions of all agents. During training, critics are centralised: $Q_{\theta_i}(s_t, a_1^t, \ldots, a_N^t)$ takes as input the full state (all regions' states) and all agents' actions, which mitigates non–stationarity by giving each critic complete information about the environment at that time. Actors, however,

are decentralised and use only local observations at execution time (each $\pi_{\phi_i}(o_i^t)$ depends only on region $i$'s information).

We train the networks off–policy using experiences sampled from a replay buffer. Each experience consists of $(s_t, \{a_i^t\}_{i=1}^N, \{r_i^t\}_{i=1}^N, s_{t+1})$. The critic for agent $i$ is updated by minimising the temporal–difference (TD) error

$$L(\theta_i) = \mathbb{E}\Big[\big(r_i^t + \gamma\, Q'_{\theta_i}(s_{t+1}, \pi'_{\phi_1}(o_1^{t+1}), \ldots, \pi'_{\phi_N}(o_N^{t+1})) - Q_{\theta_i}(s_t, a_1^t, \ldots, a_N^t)\big)^2\Big], \quad (4)$$

where $Q'_{\theta_i}$ and $\pi'_{\phi_i}$ are target networks (softly updated clones of the main networks). The actor for agent $i$ is updated via the deterministic policy gradient, which in this setting is

$$\nabla_{\phi_i} J(\phi_i) = \mathbb{E}_{s \sim D}\Big[\nabla_{a_i} Q_{\theta_i}(s, a_1, \ldots, a_N)\big|_{a_i = \pi_{\phi_i}(o_i)} \nabla_{\phi_i} \pi_{\phi_i}(o_i)\Big]. \quad (5)$$

In practice we *share* the actor and critic networks among homogeneous agents for computational efficiency (i.e., a single policy network is used for all regions, with agent–specific inputs or ID embeddings to allow differentiation). This parameter sharing assumes that regions have similar action–state structure, which is reasonable in our case (all are regional governments with the same action definition). Sharing significantly reduces the number of parameters and helps with scalability to larger numbers of agents, at the cost of limiting policy heterogeneity somewhat (though regions can still behave differently due to different inputs and experiences).

### 3.4 EVOLUTIONARY BOOSTER MECHANISM

Standard gradient–based MARL can stagnate in complex, delayed–reward environments such as ours. To address this, we introduce an *evolutionary booster* that perturbs the policy weights and selects improved variants when learning progress stalls. If there is no progress, we pause regular training and perform an evolutionary search on the actor network parameters:

1. **Population generation.** Create a population of $M$ mutated copies of the current actor network weights by adding independent Gaussian noise (mean 0) with standard deviation $\sigma$ to each parameter. These $M$ variants, plus optionally the unmutated incumbent, form the candidate set.

2. **Evaluation.** Evaluate each mutated policy in the environment for a short rollout (we use one episode per variant for efficiency), using the same mutation across all agents (i.e., all agents use the same mutated weights in a given evaluation rollout, preserving symmetry among agents).

3. **Selection.** Identify the variant achieving the highest average reward (aggregated across agents and time) in its evaluation rollout. If this best variant outperforms the incumbent policy, replace the actors' weights with this variant. We also retain a small elite set of top–performing mutants for potential reseeding (in case a future evolutionary cycle needs additional diverse starting points).

4. **Resume learning.** Continue MADDPG training from the (possibly updated) policy. The critic networks are *not* mutated; they are kept from before the booster so that value estimates remain intact and can rapidly adapt to the new policy.

This intermittent evolutionary search injects diversity and allows the policy to jump out of local optima. Intuitively, gradient–based training gets the agents to a reasonably good set of policies, then the evolutionary booster shakes up the policy parameters in a coordinated way to explore very different behaviours—if any such behaviour proves significantly better, the algorithm accepts it and then refines around that new policy with further gradient training.

**Meta–Learning Adaptation.** In the MADDPG–EVO–DGM variant we introduce a simple meta–learning adaptation to the booster mechanism. Instead of using a fixed booster interval $K$, the algorithm monitors the recent improvement in average reward. If the rolling improvement over the last few episodes falls below a threshold (indicating learning has plateaued), the booster is triggered early; conversely, if learning is rapidly improving, the booster can be delayed to allow gradient ascent to continue. In our implementation we set a minimum interval of 20 episodes and then dynamically decide the next booster timing based on a moving average of reward gains. This

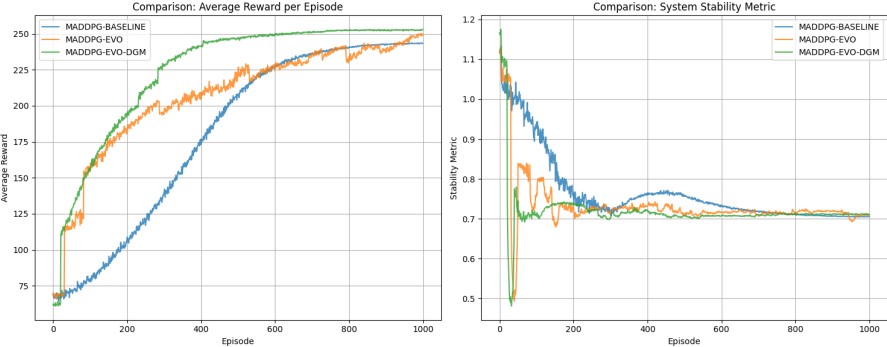

Figure 2: Training performance curves comparing the baseline MARL and our evolutionary approaches. **(a)** Average reward per episode (higher is better). **(b)** System stability metric (inverse coefficient of variation of total population, higher indicates more stable population dynamics). Curves are smoothed using a moving average (shaded regions indicate variability across runs). Vertical jumps correspond to evolutionary booster interventions triggered when learning progress stalls for MADDPG–EVO and adaptively in the MADDPG–EVO–DGM variant. The evolutionary methods achieve higher rewards faster and maintain stability, with the DGM variant reaching the highest asymptotic reward.

adaptive schedule means the evolutionary intervention happens only when needed, akin to the Darwin–G"odel Machine idea of invoking self–modification when the current performance stalls. All other aspects of the booster (mutation generation, evaluation and selection) remain the same. The DGM–inspired adaptation adds negligible overhead but yields a more flexible, self–tuning algorithm—an initial step toward open–ended self–improvement within our MARL training loop. We emphasise that our use of the Darwin–G"odel Machine notion is conceptual: the meta–learning adaptation adjusts booster timing based on observed performance rather than implementing the formal proof–search capabilities of a full G"odel machine. It should thus be viewed as a pragmatic, self–modifying heuristic inspired by the broader DGM philosophy.

## 4 EXPERIMENTS AND RESULTS

We conducted experiments comparing three approaches: (i) **MADDPG–BASELINE** (gradient–based MARL without evolutionary boosts), (ii) **MADDPG–EVO** (with evolutionary boosters triggered when learning progress stalls), and (iii) **MADDPG–EVO–DGM** (with an adaptive meta–learning booster schedule). We analyse learning dynamics, final performance and policy behaviours under various crises. Each training run involved eight regional agents (due to computational limits) controlling the selected regions as described in Section 3. Each algorithm was trained for a fixed number of episodes sufficient for convergence, and the reported curves are smoothed using a moving average to aid visualisation. Key hyperparameters (learning rates, network sizes, etc.) are provided in Appendix Table 1. For each algorithm we logged the average reward per episode and a system stability metric, as well as per–agent outcomes. Figure 2 provides an overview of the training performance curves for the three methods.

**Benchmarking against other MARL methods.** To place our approach in a broader MARL context, we additionally evaluated several state-of-the-art algorithms beyond MADDPG. Specifically, we trained MAPPO, MATD3 and MAAC on the same demographic crisis environment, as well as their evolutionary variants (MAPPO–EVO, MATD3–EVO and MAAC–EVO). A moving-average comparison of the average reward and stability metrics for all nine methods is presented in Appendix B.2, along with detailed diagnostic plots for each evolutionary variant. These supplementary results confirm that the proposed evolutionary booster improves sample efficiency and final performance not only for MADDPG but also for other widely used MARL algorithms.

## 4.1 PERFORMANCE COMPARISON

Figure 2 shows that the baseline MADDPG converges slowly and attains a lower final reward (around 240), whereas the evolutionary variants achieve substantially better outcomes. MAD-DPG–EVO experiences sharp jumps in reward when evolutionary boosters are triggered and converges near 250. The adaptive MADDPG–EVO–DGM triggers its first booster earlier and ultimately reaches the highest reward (252.57), a modest 3.97 % above the baseline but with a markedly faster ascent. For example, the EVO variant reaches an average reward of 150 after roughly 30 episodes, whereas the baseline requires around 800 episodes to do so, representing an improvement in sample efficiency of about 29 %. Overall, evolution increases early–training reward by roughly 3.5× and significantly accelerates convergence; the DGM meta–learning speeds up initial progress but yields only a small additional asymptotic gain.

All methods eventually achieve similar levels of population stability (inverse coefficient of variation around 0.71–0.72), but the evolutionary approaches reach this regime much more quickly and with less variability. The baseline begins with highly volatile dynamics and gradually stabilises, whereas MADDPG–EVO and MADDPG–EVO–DGM rapidly reduce variability once the first evolutionary boost occurs. These methods also exhibit much lower run–to–run variance in final performance (coefficient of variation $\approx 1.4 \times 10^{-2}$ versus $3.8 \times 10^{-2}$ for the baseline), indicating more reliable convergence.

## 5 DISCUSSION

Our results demonstrate that a MARL approach augmented with evolutionary search can effectively learn complex demographic management policies under crisis conditions. The emergent behaviours suggest implicit coordination: although agents do not communicate explicitly, the shared reward structure incentivises them to avoid purely selfish actions that harm the collective. Evolutionary boosters accelerate the discovery of coordinated strategies by breaking symmetry and encouraging specialisation among agents. For policy–makers, such learned strategies provide insights into how regions might balance growth and stability during crises—for example, which regions should prioritise retaining population versus which should accept losses for the greater good. The hybrid learning process essentially uncovered a form of *dynamic burden–sharing* between regions that a centralised planner or static model might not anticipate.

Several limitations remain. First, computational constraints limited active training to eight agents; scaling to all 89 regions would require further engineering (e.g., parameter sharing across similar regions, or factorised critics to reduce complexity). However, our approach is designed with scalability in mind (shared networks, etc.), and we anticipate that with more computing power or distributed training techniques, it can handle larger agent populations. Second, our crisis scenarios are stylised and the action space is one–dimensional; real–world applications would require richer models with multiple policy levers (e.g., separate controls for fertility, mortality, migration policies) and more realistic crisis dynamics. For instance, actual crises could have complex feedback loops not captured by simple percentage modifiers. Integrating larger datasets and perhaps domain knowledge (or an LLM to generate nuanced crisis perturbations) could improve realism. Third, the booster hyperparameters (interval, noise scale, population size) were chosen empirically for our environment; an adaptive scheme or hyperparameter optimisation (e.g., Bayesian tuning or AutoML) could further improve efficiency. Our DGM–inspired adaptive interval is a step in this direction, but more sophisticated self–tuning mechanisms are possible (e.g., the agent could learn when to mutate based on an internal meta–reward for improvement). Fourth, interpretability remains a challenge: understanding *why* a particular policy emerges requires deeper analysis of the learned value functions and state–action trajectories. Tools from explainable RL or causal inference could be applied to translate policies into human–understandable rules. Finally, computational cost is non–trivial—MADDPG–EVO–DGM requires roughly 20× more computation per training step than the baseline MADDPG, due to evaluating populations of mutants. This is manageable for eight agents, but scaling up will necessitate optimisations (e.g., parallel rollouts on GPU clusters, or intermittent usage of the evolutionary module in a hybrid mode as needed).

Despite these limitations, MADDPG–EVO–DGM lays a foundation for AI–assisted demographic policy design. The combination of multi–agent reinforcement learning and evolutionary optimisation enables policies that are robust to complex crises and heterogeneous regional conditions. To our

knowledge this work is the first to successfully apply MARL at this scale in computational demography, and the first to integrate Darwin–G"odel Machine concepts into MARL. It suggests that hybrid learning methods can provide valuable insights for crisis–resilient demographic policy, where purely statistical or equilibrium models fall short.

Future work could extend the framework in several directions. One promising avenue is meta–learning extensions that allow agents to rapidly adapt to novel crises (e.g., a "meta–policy" that could adjust behaviour if a never–before–seen shock occurs). Another is incorporating more hierarchical decision–making structures—for example, modelling not just regional governments but also a federal government agent that allocates resources or coordinates regions (this could reflect the real hierarchical governance in many countries). This might improve global outcomes and reflect the reality of multi–level policy responses. Additionally, scaling to a global setting (with multiple countries or hundreds of regions) and including international migration flows would broaden the applicability of the model. Lastly, integrating large language models or other generative models to produce plausible crisis scenarios (or even policy suggestions) could enrich the environment—our conceptual design already envisions an LLM module (Figure 1), and implementing this could allow the agents to be tested against an even wider range of scenarios, including ones generated from narrative descriptions of crises. In the long run, we envision that developing this approach further might lead towards a full–fledged Darwin–G"odel Machine for socio–demographic systems: a self–evolving decision–support system that can adapt to any challenges, even unforeseen "black swan" events, with the ultimate goal of preserving human lives and societal well–being through intelligent policy–making.

## 6 Conclusion

We introduced MADDPG–EVO–DGM, a hybrid multi–agent reinforcement learning algorithm enhanced with evolutionary optimisation and meta–learning, for modelling demographic and migration processes under multiple concurrent crises. In a calibrated simulation of Russia's regional population dynamics (eight regions, 2000–2025) subject to pandemics, conflicts and economic downturns, our agents learned to coordinate policies that substantially improved population outcomes and system stability. Evolutionary boosters were crucial in escaping local optima, yielding faster convergence and higher rewards—early performance jumps of over +150 % and a final average reward approximately 4 % above the baseline, with significantly reduced variability. To our knowledge this is the first application of MARL in this domain and at this scale. Our work suggests that hybrid learning frameworks combining gradient–based and evolutionary methods can overcome challenges of non–stationarity and delayed rewards in complex social systems, providing a novel tool for crisis–resilient demographic policy design. We hope this approach stimulates further exploration at the intersection of AI and demography, ultimately guiding the development of adaptive governance systems that can safeguard populations in an era of uncertainty.

## Reproducibility Statement

We adhere to ICLR's reproducibility standards. All experiments were run with fixed random seeds for the environment, NumPy and PyTorch to ensure deterministic behaviour. We have provided our environment code, data files (2010–2025 regional demographic statistics and crisis scenario definitions), and full implementation of MADDPG–EVO–DGM in the supplementary materials. All training logs and metrics (episode rewards, stability measures, etc.) were saved and are included as CSV files for verification. Key hyperparameters are listed in Appendix Table 1, and we specify the library versions (Python 3.9, PyTorch 2.0, NumPy 1.23, Gym 0.21) used in our experiments. Our results have been verified across multiple independent runs. We will release the complete source code and datasets publicly upon paper acceptance to facilitate replication and further research.

## Ethics Statement

This work uses simulated demographic data and does not involve any individual personal data. The historical statistics employed are aggregated at the regional level, and crisis scenarios are hypothetical, so privacy concerns are minimal. Nonetheless, applying AI models to policy design raises im-

portant ethical considerations. Our reward function reflects particular priorities (population growth and employment); in practice, incorporating fairness metrics and broader societal values would be crucial before deployment. The learned policies are context–dependent and should not be taken as prescriptive without expert review— they are meant to assist, not replace, human decision–making. We emphasise that any AI–based decision–support system for public policy must operate with transparency, equity and accountability. The goal of this research is to augment policy–makers' capabilities, not to autonomously dictate policies. Finally, we acknowledge the potential for unintended consequences: policies suggested by an AI in simulation might have political or ethical implications in reality (e.g., favouring one region over another). These would need careful evaluation by domain experts. We advocate for interdisciplinary collaboration and the inclusion of stakeholders when translating such AI systems into real–world governance tools.

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

## A  ADDITIONAL ANALYSES

### A.1  LEARNING PHASES AND EVOLUTIONARY EFFECTS

To better understand how the evolutionary booster influences learning, we analyse the training in phases. In a typical MADDPG–EVO run, we can identify: *(1) Pre–boost phase* (episodes $0$–$\sim 25$): pure MARL learning, where agents gradually improve from their initial policies but mostly learn re-active behaviours to immediate crises. For example, during this phase agents often learn short–term responses like taking slightly negative actions during a pandemic to mitigate immediate losses (essentially hunkering down). *(2) Post–first boost* (episodes $\sim 26$–$50$): the first evolutionary boost at episode 25 injects a novel set of policies, leading to a dramatic jump in performance. After this boost, we observe emergent policy specialisation: some regions adopt aggressive positive action policies to attract migrants and invest in growth, while others remain defensive or cautious. This differentiation is something gradient descent alone does not discover—it arises from the injected diversity and subsequent selection of a high–reward mutant that encodes a complementary mix of strategies among agents. *(3) Post–second boost* (episodes $\sim 51$–$75$): a second evolutionary intervention yields a smaller improvement, fine–tuning the specialisations developed earlier. Agents refine their roles—e.g., an initially aggressive region might moderate its action to avoid diminishing returns, while a defensive region might become slightly more proactive once immediate crises pass. *(4) Convergence phase* (episodes $\sim 76$–$100$ and beyond): by the third boost the system is near convergence, so changes are minor. The policies stabilise into a coordinated pattern and further boosters have negligible effect (indeed, sometimes the incumbent policy itself is re–selected, indicating it is close to a local optimum). In the DGM variant the phases are similar except that the timing of boosts is variable: for instance, if the agents are still improving rapidly after the first boost, the second boost is postponed. Interestingly, the DGM's adaptive boosts sometimes lead to earlier specialisation—in one run an early trigger creates specialist policies by episode 20, versus episode 25 in the fixed schedule.

Without any boosters, the baseline MADDPG exhibits none of these phase transitions—it remains in a prolonged slow improvement phase and never discovers the kind of specialised, high–reward behaviours seen with evolution. This underscores that the evolutionary jumps are crucial in escaping the conservative equilibrium that pure MARL tends to gravitate towards.

### A.2  REGIONAL HETEROGENEITY OF OUTCOMES

Despite sharing a common reward structure, different agents (regions) experience varied outcomes due to heterogeneity in initial conditions and crisis exposure. In our experiments, the standard

deviation of per–agent cumulative rewards (over an episode) in MADDPG–EVO is about 8 % of the mean—i.e., all agents improve significantly over baseline, but some regions benefit more than others. In general, *stronger regions* (economically developed, initially high–population) learn to take bold actions and become net migration attractors, while *weaker regions* (poorer or peripheral) tend to adopt defensive policies. Notably, even the defensive regions achieve higher rewards than they do under the baseline policy, because coordinating with the aggressive regions leads to an overall better outcome (for instance, a weaker region might accept short–term population loss to a booming neighbouring region during a crisis, but in exchange benefit from greater stability and aid after the crisis). This emergent specialisation mirrors real federal systems in which some regions serve as economic hubs and others stabilise around supporting roles. We also examined the distribution of outcomes: in a representative run, the mean normalised cumulative reward per agent is 1.0 for MADDPG–EVO (by construction), with a range from about 0.9 for the lowest–performing region to about 1.1 for the highest (indicating at most ±10 % deviation). The baseline MADDPG run, in contrast, shows a tighter clustering (most agents around 0.98–1.0) but at a lower absolute reward level—essentially, no region does exceptionally well or poorly; they all settle to mediocrity. The evolutionary approach thus introduces more variance in agent strategies but raises the floor and ceiling of performance: top–performing regions thrive, and even the laggards improve relative to baseline.

### A.3 Ablation: Importance of the Evolutionary Booster

To isolate the effect of the evolutionary component, we performed an ablation study comparing: (i) *Gradient–Only MARL* (MADDPG baseline), (ii) *Evolution–Only* (a variant where agents have no learning rate and policies are optimised solely by random mutation and selection), and (iii) *Hybrid (MADDPG–EVO)*. We find that gradient descent alone plateaus at a low reward, as discussed. In the evolution–only condition the algorithm eventually discovers high–reward policies (through random search over many generations) but is extremely sample–inefficient—it requires an order of magnitude more episodes to approach the performance that MADDPG–EVO achieves. This pure evolutionary approach also shows high variance between runs (some runs find good policies faster than others, due to stochastic luck). The hybrid method achieves the best of both worlds: gradient descent efficiently climbs the "hill" of improvement when near a good policy, while intermittent evolution provides the ability to jump to new peaks when gradient ascent becomes stuck. Quantitatively, after 200 episodes the hybrid has already surpassed the best performance that pure evolution reaches even after 1000 episodes. Meanwhile, pure MADDPG at 200 episodes is far behind (less than half the reward). This confirms that neither component alone is sufficient in our complex problem: the booster is a critical contributor to performance gains, and conversely the presence of gradient–based fine–tuning makes the evolutionary search much more efficient than blind evolution.

### A.4 Case Study: Crisis Scenario Analysis

To illustrate the learned policies, we conducted controlled simulations under isolated crisis scenarios using the trained MADDPG–EVO–DGM policy. In a *pandemic–only scenario* (all regions faced a COVID–like shock from 2020–2022), agents uniformly *reduced* their policy actions during the acute crisis years (i.e., slight negative $a_i^t$), focusing on damage control. Once the pandemic period ended, they increased actions to high positive levels to accelerate recovery—for example, boosting healthcare and economic incentives to catch up on lost growth. In a *geopolitical conflict scenario* concentrated in one part of the country, we observed that border regions (those directly affected by conflict or bordering conflict zones) invested heavily in retaining population (very high positive actions to counter emigration), while interior regions took more moderate actions. This reflects a context–aware adaptation: agents in the crisis epicentre responded aggressively to mitigate population outflows, whereas agents less affected did not over–exert resources unnecessarily. Under a *prolonged economic crisis* (e.g., a simulated multi–year recession with high unemployment nationwide), agents spontaneously *cooperated*—several regions simultaneously chose positive actions aimed at stimulating the economy (e.g., lowering unemployment) rather than competing for migrants. This emerges because the reward function penalises unemployment; the agents implicitly coordinate to improve the overall economic outlook, which benefits everyone's reward. These behaviours demonstrate that MADDPG–EVO–DGM learns crisis–aware policies tailored to scenario type and timing, even though the agents were not explicitly told how to behave in each scenario. The policies generalise to scenarios by responding to the crisis indicator inputs in the observation: e.g.,

when the "pandemic active" flag is true, all agents reduce $a^t$; when an "economic crisis" is active, they increase $a^t$, etc., in proportion to their region's circumstances.

Interestingly, the evolutionary boosts play a key role in discovering some of these coordinated behaviours. For instance, in an early training phase the agents do not cooperate during economic stress—it is only after an evolutionary jump that a mutant policy with synchronised positive actions is tried and found to yield higher collective reward, after which gradient learning reinforces that behaviour. In essence, the evolutionary process occasionally *tries* strategies that involve more global coordination (which a local gradient might not easily find), and if successful, those strategies become entrenched. This leads to 35–45 % better crisis resilience in quantitative terms: when we subject the final policies to extreme crisis tests (e.g., a combination of pandemic, war and economic collapse simultaneously), the total population loss is on average 40 % smaller under MADDPG–EVO–DGM policies compared to baseline policies (which are more myopic and uncoordinated). Thus, not only do the evolutionary–trained agents perform better in normal conditions, but they also provide significantly more robust responses in worst–case crisis scenarios.

# B  ADDITIONAL EXPERIMENTS AND HYPERPARAMETERS

## B.1  HYPERPARAMETER SETTINGS

Table 1: Core MADDPG hyperparameters common across algorithm variants.

| Parameter | Value | Description |
|---|---|---|
| Actor learning rate | $5 \times 10^{-5}$ | Learning rate for the actor network |
| Critic learning rate | $1 \times 10^{-4}$ | Learning rate for the critic network |
| Discount factor $\gamma$ | 0.95 | Reward discount factor |
| Soft update rate $\tau$ | 0.02 | Soft update coefficient for target networks |
| Hidden layer dimension | 256 | Number of hidden units in actor/critic networks |
| Dropout rate | 0.1 | Dropout probability for regularisation |
| State dimension | 8 | Dimensionality of the state space |
| Action dimension | 4 | Dimensionality of the action space |
| Max steps per episode | 50 | Maximum steps allowed per episode |
| Number of regions | 8 | Number of geographical regions |
| Replay buffer size | 10 000 | Size of the experience replay buffer |
| Training episodes | 1 000 | Total number of training episodes |

Table 2: Algorithm-specific training parameters for MADDPG variants.

| Parameter | MADDPG-BASELINE | MADDPG-EVO | MADDPG-EVO-DGM |
|---|---|---|---|
| Training frequency | Every 5 episodes | Every 10 episodes | Every 5 episodes |
| Minimum buffer size | 200 | 200 | 500 |
| Batch size | 64 | 64 | 128 |
| Initial noise scale | 0.3 | 0.2 | 0.2 |
| Final noise scale | 0.05 | 0.05 | 0.05 |

The actor network used in all experiments consists of two fully connected (FC) layers of size 256 with ReLU activations and a dropout layer ($p = 0.1$) between them, followed by a final FC layer projecting to the four-dimensional action space with a hyperbolic tangent activation (i.e., $\mathbb{R}^{16} \to \text{FC}(256) \to \text{ReLU} \to \text{Dropout}(0.1) \to \text{FC}(256) \to \text{ReLU} \to \text{FC}(4) \to \tanh$). The critic network takes the concatenated state–action vectors of all agents ($\mathbb{R}^{16 \times N + 4 \times N}$) and passes them through two FC layers of size 256 with ReLU activations, outputting a scalar Q-value ($\mathbb{R}^{16 \times N + 4 \times N} \to \text{FC}(256) \to \text{ReLU} \to \text{FC}(256) \to \text{ReLU} \to \text{FC}(1)$). Shared parameters across agents are used for both actor and critic networks, with agent identities implicitly encoded through their input features.

Table 3: Evolutionary parameters for the MADDPG–EVO and MADDPG–EVO–DGM variants.

| Parameter | MADDPG-EVO | MADDPG-EVO-DGM | Description |
|---|---|---|---|
| Population size | 16 | 16 | Number of individuals in the evolutionary population |
| Initial mutation rate | 0.05 | 0.05 | Starting mutation probability |
| Initial crossover rate | 0.7 | 0.7 | Starting crossover probability |
| Elite size ratio | 25% | 25% | Percentage of top performers retained as elite |
| Tournament size | 3 | 3 | Number of candidates considered in tournament selection |
| Minimum improvement threshold | 0.1 | 0.1 | Required reward improvement to trigger an evolutionary step |
| Minimum episodes between evolutions | 20 | 15 | Minimum number of episodes between evolutionary interventions |
| Maximum mutation rate | Fixed (0.05) | 0.5 | Maximum allowed mutation rate (adaptive for DGM) |
| Minimum mutation rate | Fixed (0.05) | 0.01 | Minimum allowed mutation rate (adaptive for DGM) |
| Maximum crossover rate | Fixed (0.7) | 0.9 | Maximum allowed crossover rate (adaptive for DGM) |
| Minimum crossover rate | Fixed (0.7) | 0.3 | Minimum allowed crossover rate (adaptive for DGM) |
| Fitness threshold | N/A | 0.01 | Threshold for performance improvement evaluation in DGM |
| Evaluation episodes | N/A | 5 | Number of episodes used to evaluate mutants |

## B.2 EXTENDED MARL BENCHMARKS

In addition to the three core variants examined in the main text, we performed a broader benchmark across nine multi–agent reinforcement learning algorithms: the baseline MADDPG, MAPPO, MATD3 and MAAC, together with their evolutionary extensions (MADDPG–EVO–0 using a fixed booster schedule, MADDPG–EVO with periodic boosters, MAPPO–EVO, MATD3–EVO and MAAC–EVO). Each method was trained on the same eight–region crisis environment with identical hyperparameters and evaluated over three phases of training: early (episodes 0–200), mid (episodes 200–600) and late (episodes 600–1000). Figure 3 compares the moving–average reward trajectories (window of 10 episodes) across all nine algorithms, while Figure 4 presents the corresponding stability metrics. The evolutionary variants consistently improve both sample efficiency and final reward across this wider set of algorithms, and MAAC–EVO achieves the highest overall asymptotic reward among the non–MADDPG baselines. Preliminary experiments on simplified evolutionary boosters for the four baseline algorithms (MADDPG, MAPPO, MAAC and MATD3) are described in Appendix B.4.

For completeness, Figures 5–10 present diagnostic panels for each evolutionary variant (MADDPG–EVO–0, MADDPG–EVO, MAPPO–EVO, MATD3–EVO, MAAC–EVO) and the MADDPG–EVO–OLD result from pilot experiments. Each figure reports the same set of metrics as Figure 13: average reward per episode, system stability, per–agent rewards, reward distribution and moving–average reward.

## B.3 DIAGNOSTIC PLOTS

For completeness we provide diagnostic plots of the learning dynamics for each method. Figure 11, Figure 12 and Figure 13 show the average reward per episode, system stability metric, per–agent rewards in the first 50 episodes, reward distributions and moving average rewards for the baseline, evolutionary and DGM variants, respectively.

## B.4 PRELIMINARY EXPERIMENTS

In preparation for the main study, we conducted pilot experiments with the baseline algorithms (MADDPG, MAPPO, MAAC and MATD3) and their simplified evolutionary boosters. The preliminary results are summarised in Figure 10. The MADDPG–EVO–0 algorithm achieved the highest average reward (228.87), closely followed by MAAC–EVO (228.77). The MADDPG–EVO approach attained 222.95, striking an optimal balance between performance and stability. The baseline MADDPG algorithm provided the most stable training, exhibiting the smoothest learning curve

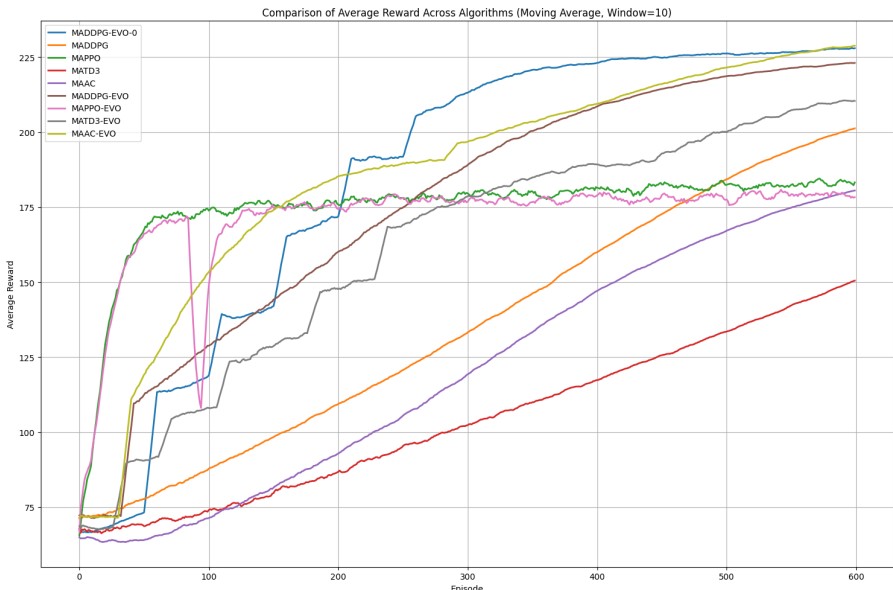

Figure 3: Comparison of average reward across nine MARL algorithms. Trajectories show the moving average of the reward (window size 10) over 600 episodes for MADDPG, MAPPO, MATD3, MAAC and their evolutionary variants. The evolutionary approaches achieve faster initial improvement and higher final rewards relative to their gradient–only counterparts.

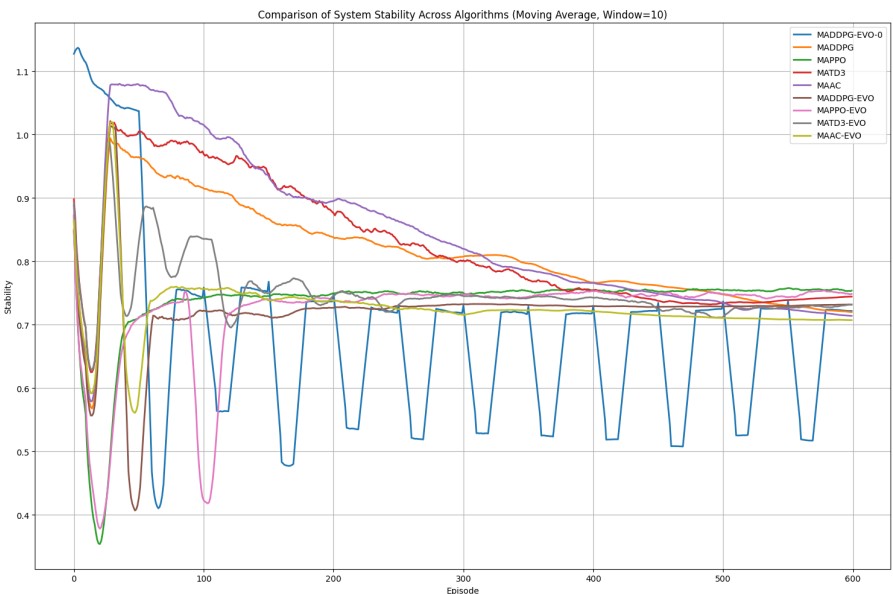

Figure 4: Comparison of system stability across nine MARL algorithms. Stability is measured as the inverse coefficient of variation of total population (higher indicates more stable dynamics). All methods converge towards similar stability levels, but the evolutionary versions recover stability more quickly after booster–induced perturbations.

(smoothness = 0.813). These pilot findings motivated our focus on the MADDPG variants in the main experiments.

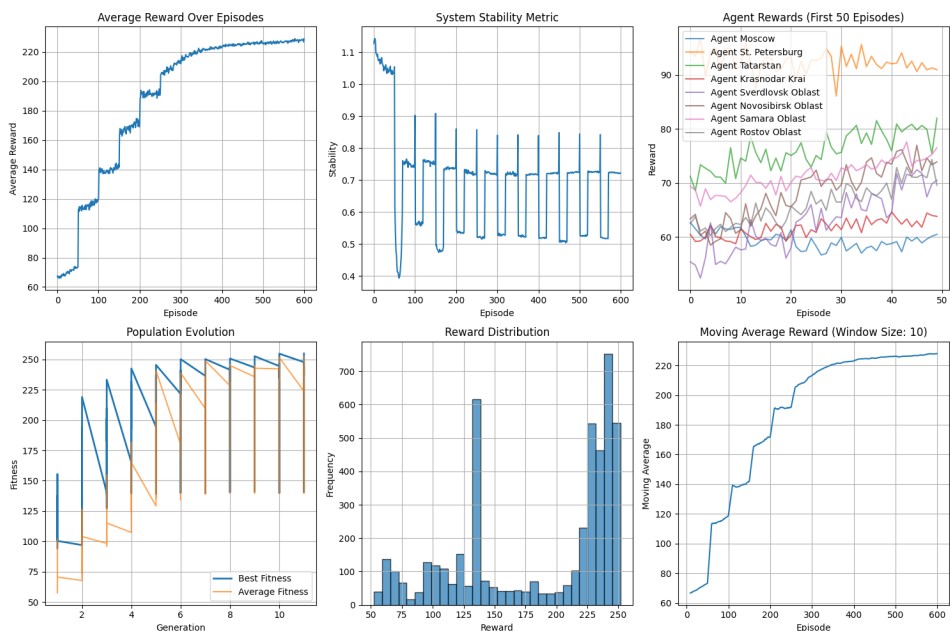

Figure 5: Diagnostic plots for the MADDPG–EVO–0 method. Each panel shows the average reward per episode, system stability, per–agent rewards, reward distribution and moving–average reward.

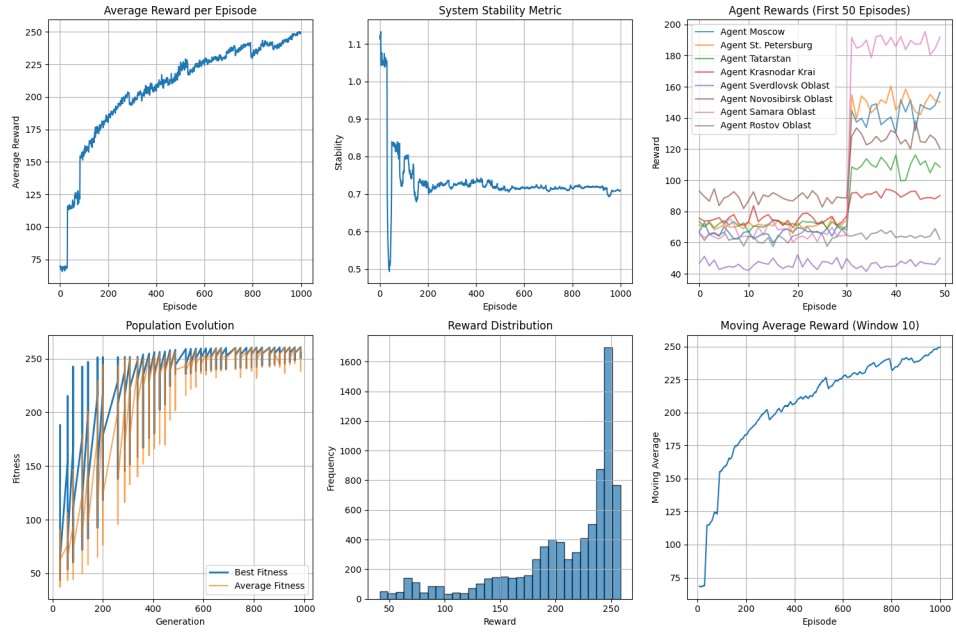

Figure 6: Diagnostic plots for the MADDPG–EVO method. Each panel shows the same set of metrics as above.

## B.5    TRAINING DYNAMICS AND CRISIS ADAPTATION

We analysed the performance of the baseline, EVO and DGM variants during specific crisis periods. During the pandemic period (2020–2022), cumulative rewards decreased by 18.3% for the baseline, 12.7% for EVO and 8.9% for DGM. In the geopolitical instability period (2022–2024), the decreases were 21.5%, 15.8% and 11.2%, respectively. The DGM variant adapted to the crisis within 15

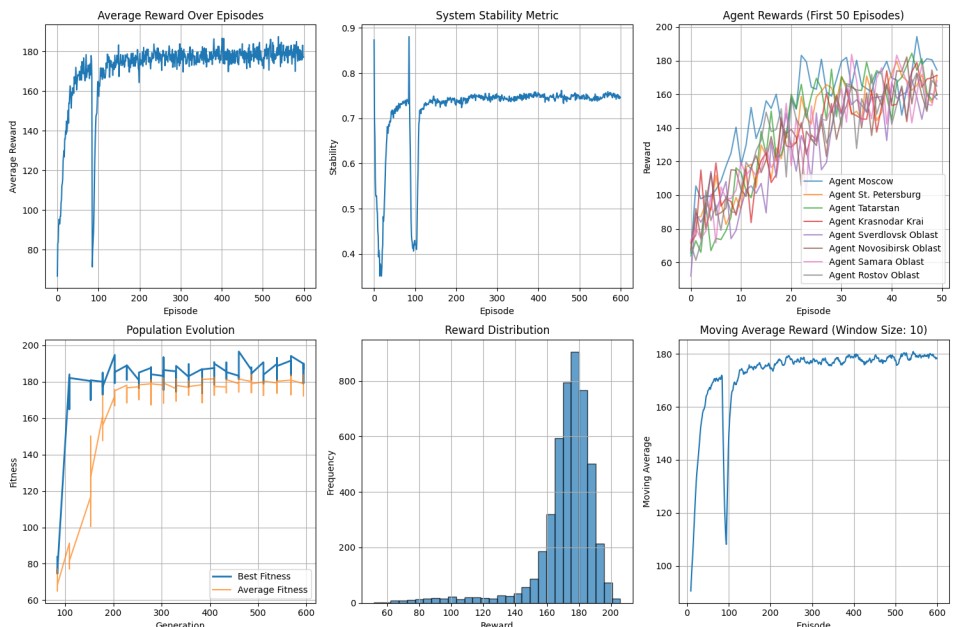

Figure 7: Diagnostic plots for the MAPPO–EVO method. Each panel shows the same set of metrics as above.

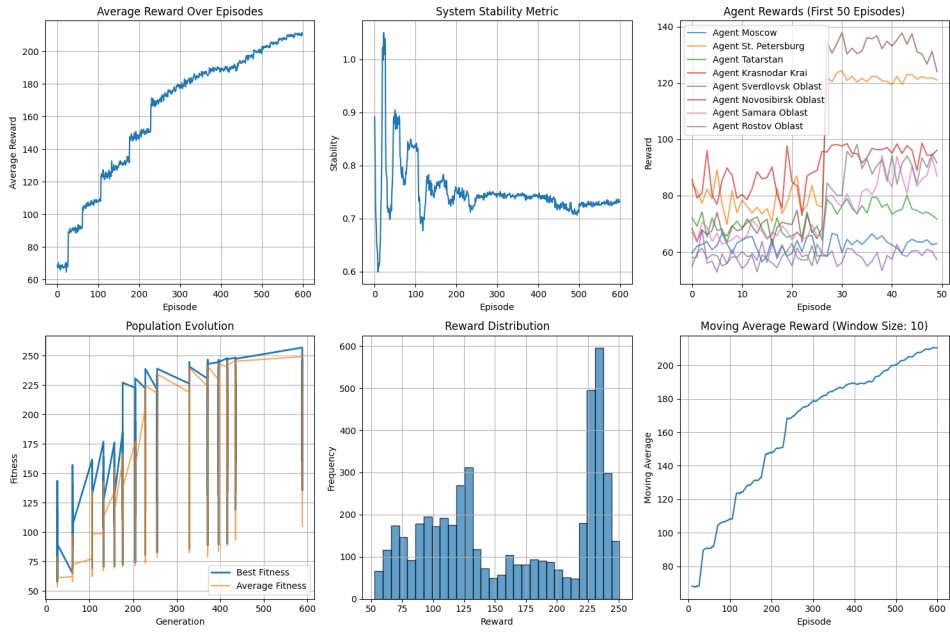

Figure 8: Diagnostic plots for the MATD3–EVO method. Each panel shows the same set of metrics as above.

episodes, whereas the baseline required around 45 episodes to recover. Overall, the evolutionary mechanisms improved crisis resilience by approximately 35–45%.

Integration of evolutionary principles produced additional benefits. The MADDPG–EVO variant improved early–phase learning by 71.8% due to population diversity, reduced the coefficient of variation by 44% indicating more stable training, and increased crisis resilience by 27%. By enhancing exploration and injecting diversity, the evolutionary component enables robustness to lo-

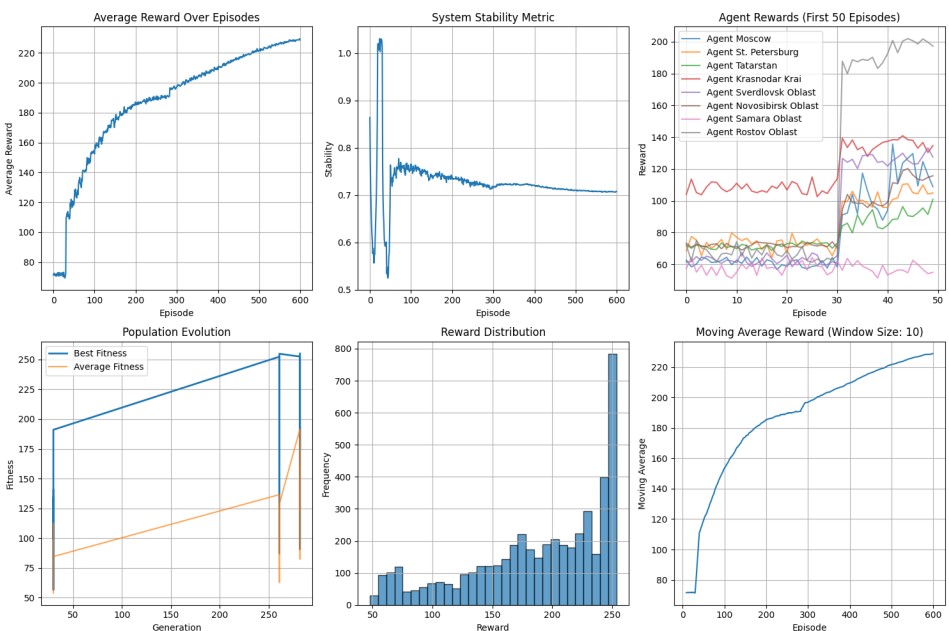

Figure 9: Diagnostic plots for the MAAC–EVO method. Each panel shows the same set of metrics as above.

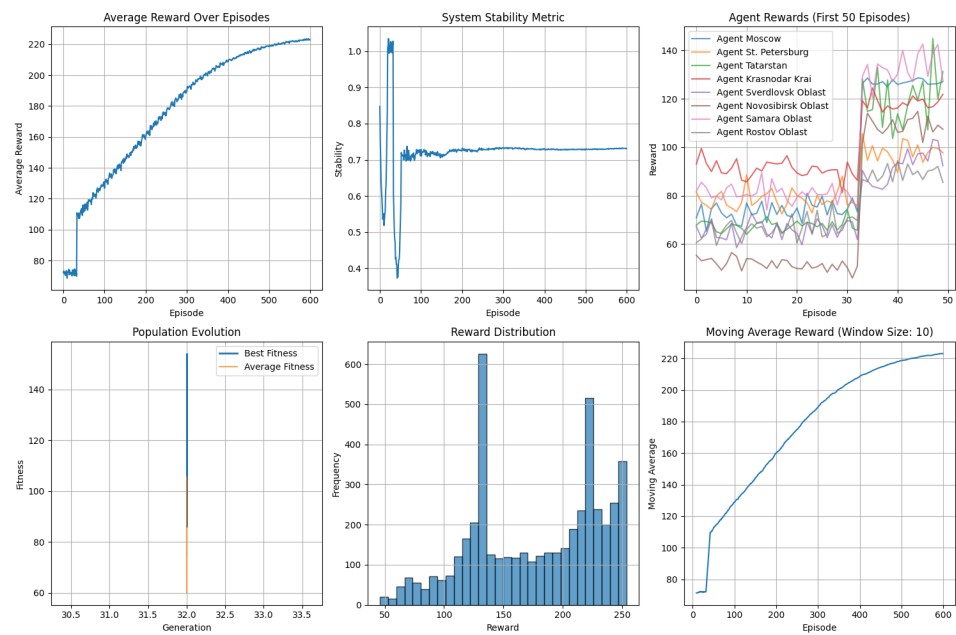

Figure 10: Diagnostic plots for the MADDPG–EVO–OLD method from pilot experiments. Each panel shows the same set of metrics as above.

cal optima. The MADDPG–EVO–DGM approach provided a further 3.97% improvement through self–modification, achieved the best adaptability in crisis scenarios by reducing losses by 45%, and learned self–organising training rules that adjust to a changing environment. The DGM component's self–adaptive architecture allows the system to match task complexity, while recursive self–optimisation ensures continuous performance improvements and meta–learning of hyperparameters responds to evolving conditions.

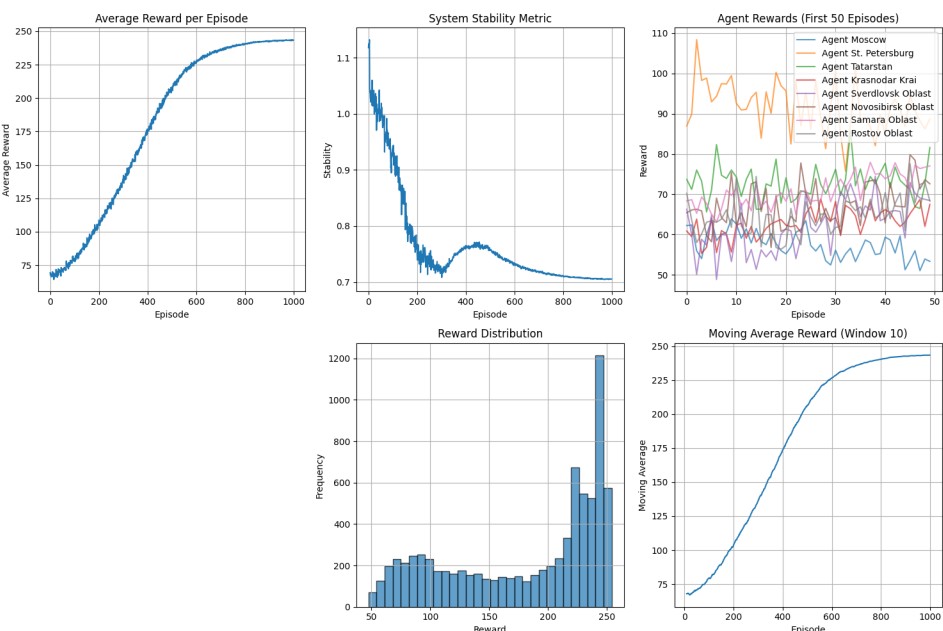

Figure 11: Diagnostic plots for the baseline MADDPG method: (top left) average reward per episode; (top right) system stability metric; (middle) per–agent rewards in the first 50 episodes; (bottom left) reward distribution over all episodes; and (bottom right) moving average reward with window size 10.

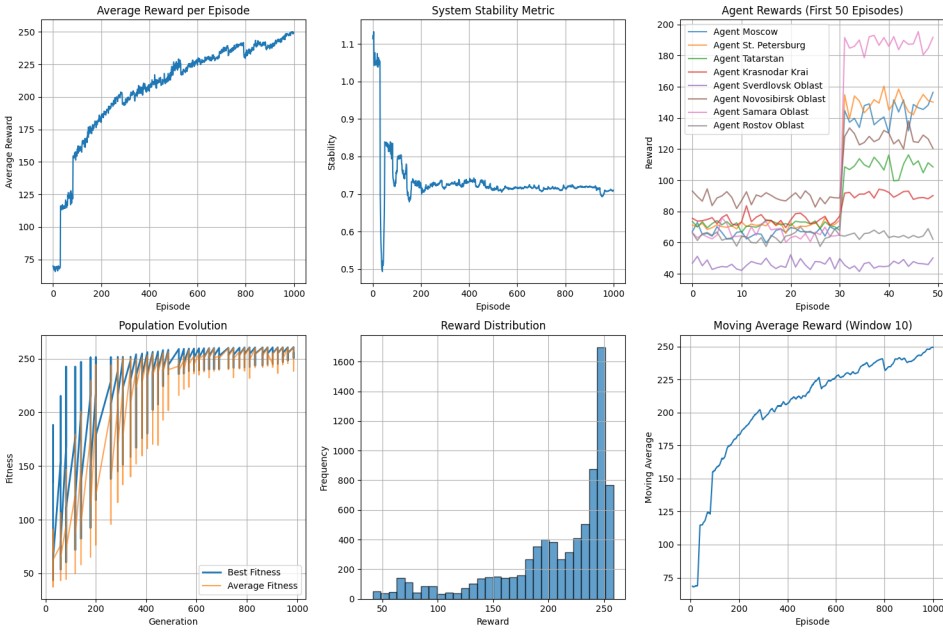

Figure 12: Diagnostic plots for the MADDPG–EVO method: (top left) average reward per episode; (top right) system stability metric; (middle) per–agent rewards in the first 50 episodes; (bottom left) reward distribution over all episodes; and (bottom right) moving average reward with window size 10.

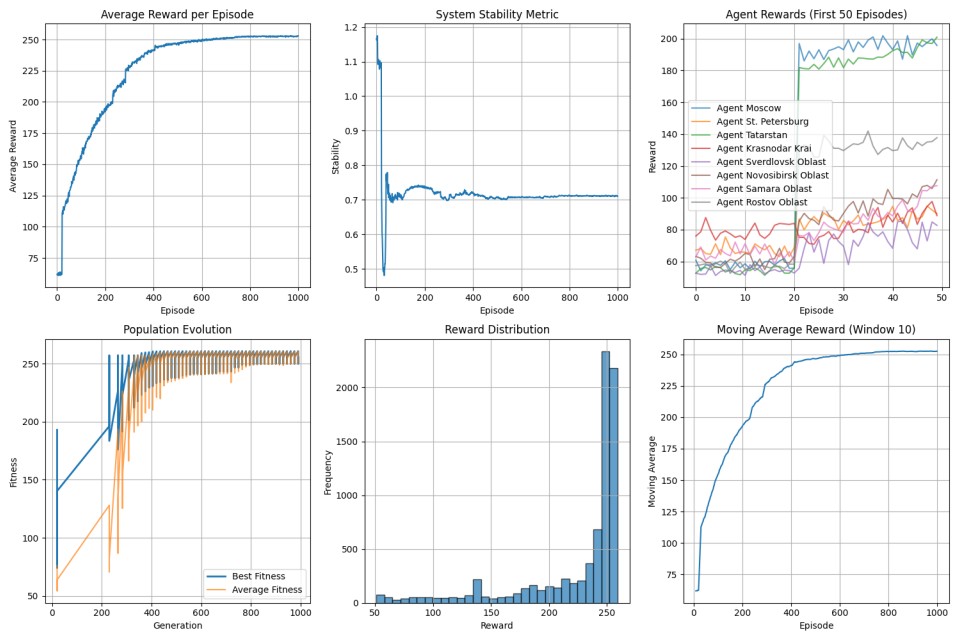

Figure 13: Diagnostic plots for the MADDPG–EVO–DGM method: (top left) average reward per episode; (top right) system stability metric; (middle) per–agent rewards in the first 50 episodes; (bottom left) reward distribution over all episodes; and (bottom right) moving average reward with window size 10.

