# OpenReview forum: "Evolutionary Multi-Agent Reinforcement Learning for Crisis-Aware Demographic Policy Optimization"
_ICLR.cc/2026/Conference — Submitted to ICLR 2026_

### Official Review · Reviewer_kGpr · 2025-10-23

**Soundness:** 2
**Presentation:** 1
**Contribution:** 2
**Rating:** 2
**Confidence:** 4

**Summary:**

The paper introduces MADDPG-EVO-DGM, a hybrid multi-agent reinforcement learning (MARL) algorithm that combines Multi-Agent Deep Deterministic Policy Gradient (MADDPG) with evolutionary optimization and meta-learning principles.
This approach is applied to large-scale demographic modeling under multiple concurrent crises, treating regions as autonomous agents that learn to optimize policy levers in a simulated environment calibrated to real data.

**Strengths:**

- Integration of MARL with demographic crisis modeling.

- Meta-learning adaptation introduces self-modification.

**Weaknesses:**

- Numerous studies [1] have employed evolutionary algorithms to tune model parameters for optimizing the training process, rendering this approach less innovative.

- The demographic environment, while incorporating multiple regions and their interactions, remains relatively simplistic. The crisis parameters are fixed, and each action spans one year, which may be excessively long for optimal control yet too brief to effectively reflect demographic policy impacts.

- The experimental results are overly simplistic, lacking comparisons with advanced baseline methods. Additionally, there is no in-depth analysis of the proposed design or the demographic scenarios explored.

- Figure 1 is confusing and appears inconsistent with the approach outlined in the paper. For instance, the use of LLMs is not referenced in Section 3. Furthermore, the distinction between the two "crisis modeling" components in the figure is unclear. Lastly, why there is an arrow pointing from 'MARL' to 'crisis modeling', rather than the reverse?

[1] Bai, Hui, Ran Cheng, and Yaochu Jin. "Evolutionary reinforcement learning: A survey." *Intelligent Computing* 2 (2023): 0025.

**Questions:**

see weaknesses.

---

### Official Review · Reviewer_qPZ9 · 2025-10-27

**Soundness:** 2
**Presentation:** 2
**Contribution:** 2
**Rating:** 2
**Confidence:** 3

**Summary:**

This paper proposes MADDPG-EVO-DGM to optimize regional demographic policies under severe crisis conditions. To address sparse rewards and suboptimal local optima, the paper introduces an evolutionary booster and Darwin-Gödel Machine based meta-learning on top of a MADDPG baseline. The method is evaluated in a custom, crisis-aware simulation environment calibrated with 25 years of real demographic data for eight federal regions of the Russian Federation and ten concurrent crisis scenarios. The results show that the proposed approach converge significantly faster, achieve a higher final average reward, and demonstrate a 3.4x lower convergence variance than the baseline.

**Strengths:**

1. The paper applies modern MARL techniques to large-scale, national demographic policy optimization, especially in the context of concurrent crises. This is a highly novel and socially significant problem, and the paper provides a strong proof-of-concept.
2. The authors have clearly invested significant effort in building a simulation environment that is both complex and grounded.
3. The core idea of an evolutionary booster is an effective solution to the exploration problem. The results in Figure 2 compellingly demonstrate that this hybrid approach is crucial for success.

**Weaknesses:**

1. The entire complexity of regional demographic policy is compressed into a single, continuous 1D action ("abstract policy lever"). This is a massive simplification. Real-world policy involves a high-dimensional, complex action space (e.g., separate budgets for healthcare, migration programs, infrastructure). It is unclear if the learned policies are truly meaningful or just an artifact of this 1D control problem.
2. The MADDPG-EVO-DGM method requires roughly 20x more computation per training step than the baseline MADDPG. This enormous cost makes the method's practical scalability, especially to the 89 regions motivated in the introduction, highly questionable.
3. Figure 1 is confusing.
4. The improvements over baselines are not very significant.

**Questions:**

1. How to validate the fidelity of the simulation?
2. See weaknesses.

---

### Official Review · Reviewer_nUvP · 2025-10-31

**Soundness:** 3
**Presentation:** 2
**Contribution:** 2
**Rating:** 6
**Confidence:** 4

**Summary:**

The paper presents MADDPG–EVO–DGM, a new method based on multi-agent reinforcement learning and meta-learning, applied to demographic policy design and crisis handling. MADDPG–EVO–DGM augments the existing MADDPG algorithm with periodic evolutionary boosters modulated via meta-learning. The method is studied on a novel simulation environment based on real-world demographic data, finding that MADDPG-EVO-DGM: (i) performs moderately better than MADDPG with significantly faster and more stable convergence; and (ii) significantly more robust than MADDPG in extreme crisis scenarios.

**Strengths:**

Quality:
- the claim that MADDPG-EVO-DGM improves over MADDPG is supported by a direct, experimentally sound comparison that finds MADDPG-EVO-DGM to have moderately increased performance with significantly better sample efficiency and training stability
- the importance of evolutionary boosters is further validated on a wide set of MARL algorithms in the appendix
- reproducibility is supported by the supplementary material, which provides hyperparameters and code

Originality and significance:
- while the contribution is specific to the presented domain (demographic policy design), it still presents a new method that combines existing MARL, evolutionary and meta-learning methods in a meaningful way
- the presented method, simulation environment and results, both quantitative and qualitative (including the supplementary material) are a meaningful contribution to demographic policy design

Clarity:
- overall, the manuscript is clear and the layout of the content is predictable
- the contribution of the paper is clear and properly contextualized both in the introductory and related work sections
- the idea of employing adaptive evolutionary boosters to avoid local minima is intuitive and sound

**Weaknesses:**

Quality:
- while the experimental setup is described in adequate detail, the number of runs used to compute the variance of the training curves does not seem to be reported. Since this detail can greatly affect the significance of the corresponding results, especially in (MA)RL, it should be reported in the main text, ideally in the caption of the figures showing the convergences curves
- it is mentioned that only 8 regions where simulated due to computational constraints, but there is no discussion on why this is the case (e.g. identifying bottlenecks), and how/if this would change in real-world applications

Originality:
- the novelty of MADDPG-EVO-DGM lies mainly in the adaptiveness/sparse frequency of the "evolutionary boosters", i.e. it is mostly an original combination of existing methods

Significance:
- while MADDPG-EVO-DGM is novel as a method, it is only applied to an arguably niche MARL domain (demographic policy design) on a novel environment, preventing this contribution from being relevant for other domains to which MARL is applicable
- while it is clear that evolutionary boosters improve existing MARL methods, the results lack a direct comparison with previous non-MARL methods for demographic modeling, or a proper discussion (in the section that discusses experimental results) of why they would not be applicable at all to the proposed experimental setup (if this is the case). This would greatly enhance the significance by complementing the differences discussed in the related work section with direct evidence.

Clarity:
- the description of the environment can be improved, especially for readers that are not necessarily familiar with demographic modeling. For example, it might not be immediately clear why an agent that only optimizes the given reward (eq. 2) would not be quickly drawn to perform very large actions, since there seems to be no explicit penalty for it. Discussing this, ideally with a simple example, would better motivate the use of MARL in this domain

**Questions:**

- how many runs where used to compute the average and deviation of the training curves in all experiments?
- are previous non-MARL methods for demographic modeling applicable to the same experimental setup? How do they compare against MARL and MADDPG-EVO-DGM?
- what exactly makes the problem of demographic policy design complex enough to require being modeled as a full Markov game to be solved with general-purpose (meta-)MARL algorithms?

---

### Meta-Review · Area_Chair_iQVM · 2026-01-11

**Summary:**

This submission proposes MADDPG-EVO-DGM, a hybrid multi-agent RL approach for crisis-aware demographic policy optimization. One reviewer is marginally above accept (noting sound comparison and stability gains), while two recommend reject due to limited novelty beyond combining known components, weak/unclear experimental grounding, simplified environment/action design, compute cost, and presentation issues. The author did not provide response. The AC agrees with the majority of the reviewers on rejecting the paper.

**Reviewer Concerns:**

see Summary

**Reviewer Scores:**

see Summary

---

### Decision · Program_Chairs · 2026-01-26

Reject